# DriVLMe: Enhancing LLM-based Autonomous Driving Agents with Embodied and Social Experiences

Yidong Huang[1]    Jacob Sansom[1]    Ziqiao Ma[1*]    Felix Gervits[2]    Joyce Chai[1]
[1]University of Michigan    [2]Army Research Lab
https://sled-group.github.io/driVLMe/

## Abstract

*Recent advancements in foundation models (FMs) have unlocked new prospects in autonomous driving, yet the experimental settings of these studies are preliminary, over-simplified, and fail to capture the complexity of real-world driving scenarios in human environments. It remains under-explored whether FM agents can handle long-horizon navigation tasks with free-from dialogue and deal with unexpected situations caused by environmental dynamics or task changes. To explore the capabilities and boundaries of FMs faced with the challenges above, we introduce DriVLMe, a video-language-model-based agent to facilitate natural and effective communication between humans and autonomous vehicles that perceive the environment and navigate. We develop DriVLMe from both embodied experiences in a simulated environment and social experiences from real human dialogue. While DriVLMe demonstrates competitive performance in both open-loop benchmarks and closed-loop human studies, we reveal several limitations and challenges, including unacceptable inference time, imbalanced training data, limited visual understanding, challenges with multi-turn interactions, simplified language generation from robotic experiences, and difficulties in handling on-the-fly unexpected situations like environmental dynamics and task changes.*

## 1. Introduction

Autonomous driving (AD) has made remarkable progress in recent years, bringing us closer to a future where vehicles can function as our social robot partners that navigate roads safely and efficiently with minimal human intervention [42, 56]. As these AD agents start to enter our everyday lives, techniques to enable effective human-agent dialogue and collaboration become important. The ability to communicate with humans through natural language dialogue plays a crucial role in ensuring passenger safety, re-

covering from unexpected situations, gaining trustworthiness, and enhancing the overall driving experience [26, 60]. In traditional autonomous driving systems and in-vehicle dialogue systems, rule-based approaches [1, 35, 41] have been employed to interpret human instructions and generate appropriate responses. However, these systems often struggle to handle the complexity and variability of natural language, leading to limited functionality and sub-optimal performance. Recently, the paradigm has shifted to data-driven learning-based approaches [5, 14, 17, 19], which offer language-based interpretability and promising results in short-horizon tasks.

Advances in foundation models (FMs) like Large Language Models (LLMs) have opened up new opportunities, as they demonstrate the ability to perform step-by-step reasoning [58], to understand multimodal data [66, 69], to learn from embodied experiences [32, 61], and to use external tools [40]. An increasing number of efforts [18, 25, 43, 45, 50, 59, 62] have demonstrated the potential of FMs in the field of autonomous driving. However, the experimental setups of these works are preliminary and simplified, compared to the real driving scenarios in human environments. One common limitation is the lack of an ability to handle long-horizon navigation tasks. Trained on simple action-level natural language instructions, these models perform well on short-horizon tasks like *turn* or *overtake* but fail to understand goal-level instructions that require route planning and map knowledge. Also, these systems only focus on following individual instructions in a single turn of interaction. Realistic interactions with human passengers often involve free-form dialogue, especially for collaboratively handling unexpected situations, e.g., those caused by sensor limitations, environmental dynamics, or task changes. Without modeling the interaction context, these models may fall short of understanding nuanced dialogue and providing appropriate responses in human-vehicle interactions.

To explore the capabilities and boundaries of FMs faced with the challenges above, we introduce DriVLMe, a novel video-language-model-based AD agent to facilitate natural and effective communication between humans and au-

---

*Correspondence, contact: marstin@umich.edu

tonomous vehicles that perceive the environment and navigate. Motivated by Hu and Shu [15], our goal is to enhance a language model backend as world and agent models. We develop DriVLMe by learning from both *embodied experiences* in a simulated environment and *social experiences* from real human dialogue. Unlike previous works that only focus on open-loop benchmark evaluation using non-interactive datasets such as nuScenes [3] and BDD [65], we present both open-loop and closed-loop experiments in a simulated environment (i.e., CARLA [9]). For open-loop evaluations, we leverage the Situated Dialogue Navigation (SDN) [26] and the BDD-X [20] benchmarks to assess DriVLMe's performance in generating dialogue responses and physical actions. Our experimental results have shown that DriVLMe significantly outperforms previous baselines on SDN by a large margin and competes with baselines trained with LLM-augmented data. We further conduct closed-loop pilot studies in the CARLA simulation environment. DriVLMe is engaged in dialogue to follow language instructions from human subjects in the CARLA environment. Our preliminary findings have demonstrated some promising abilities of DriVLMe in navigation and re-planning, and on the other hand also revealed several limitations including unacceptable inference time, imbalanced training data, and low image input resolution. It remains a challenge to support multi-turn interactions and language generation from robotic experiences. We hope this paper offers a comprehensive perspective view of the strengths and weaknesses of foundation models as AD agents, highlighting areas that need future enhancement.

## 2. Related Work

### 2.1. Foundation Models for Autonomous Driving

Recent research has explored the potential of LLMs in autonomous driving, e.g., by prompt engineering on off-the-shelf LLMs to obtain the driving decisions from textual descriptions of the surrounding environment [43, 44, 59], or by fine-tuning LLMs to predict the next action or plan future trajectories [4, 29]. To develop multimodal systems, both real and simulated driving videos have been utilized for instruction tuning [47]. For example, DriveGPT4 [62] and RAG-Driver [67] fine-tuned multimodal LLMs on real-world driving videos to predict future throttle and steering angles. DriveMLM [57] and LMDrive [45] adopted camera data and ego-vehicle states from the CARLA simulator. We refer to recent surveys and position papers for detailed reviews [6, 11, 23, 63]. We note that the experimental setups in these efforts are preliminary and simplified, compared to the real driving scenarios in human environments. First, these prior approaches were restricted to single human instructions (or even no language input), limiting performance on longer-horizon tasks with back-and-forth di-

alogue and higher-fidelity navigation goals. Furthermore, these prior models only focus on using LLMs to predict physical actions and give explanations, ignoring their potential to initiate dialogue and generate language responses from robotic experiences. Finally, none of these setups consider unexpected situations caused by sensor limitations, environmental dynamics, or plan changes.

### 2.2. Language-guided Autonomous Driving and Outdoor Vision-Language Navigation

Situated human-vehicle communication has been extensively studied in the form of spoken language, and this line of work dates back to early resources including several multilingual [52] and multimodal [8, 21] speech corpora. Recently, vision-and-language navigation (VLN) tasks require an agent to navigate in a 3D environment based on natural-language instructions and egocentric camera observations, with some efforts in the outdoor scenarios [22, 53]. They consider the world as a discrete graph while agents navigate toward the goal by moving among nodes. Thanks to open-world autonomous driving simulators [9, 55, 70], recent work bridges the gap between discrete model prediction and continuous closed-loop control. Various language-guided autonomous driving experiments and datasets [26, 39, 48] have been developed based on these simulators.

### 2.3. Dialogue-guided Robotic Agents

Dialogue-guided agents for improving human-robot interaction have gained significant attention [30, 31]. Efforts in this field have ranged from enabling robots to adjust their plans in real-time based on human dialogue [7, 46], to seeking additional hints [34, 49], or to ask for direct human collaboration [33] for task completion. The advances of LLMs have infused new potential into these studies [12, 64]. For instance, InnerMonologue [16] investigates the use of LLMs for generating internal dialogue to assist in completing human-oriented tasks, while PromptCraft [38] explores precise prompt engineering to enhance the communication skills of robots. These developments underscore the pivotal role of foundation models as building blocks of agents to foster more effective human-robot collaboration.

## 3. Dorothie & Situated Dialogue Navigation

We set up our experiment in CARLA [9], a driving simulator for autonomous vehicles, and use the DOROTHIE framework [26] built upon it, which supports human-agent dialogue and various forms of unexpected situations. In this work, we adopt the problem definition and data from the Situated Dialogue Navigation (SDN) benchmark in [26].

### 3.1. Overview

The SDN benchmark is designed to assess the agent's capability in generating dialogue responses and physical navi-

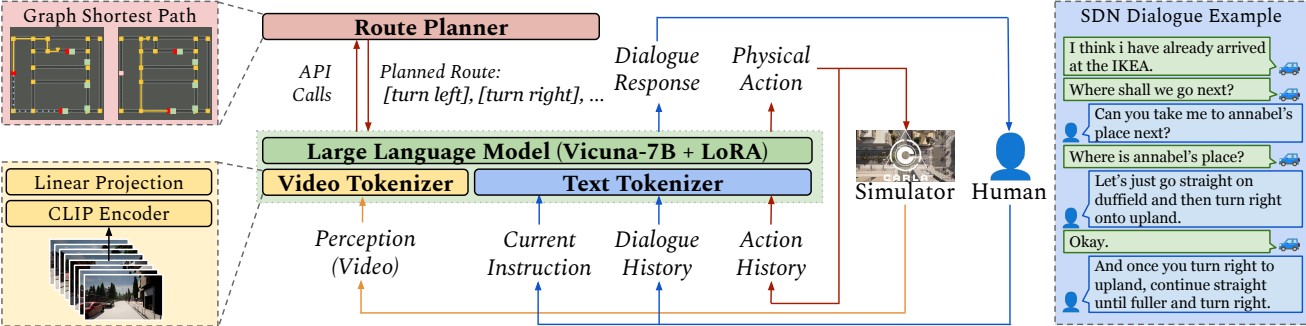

Figure 1. Overview of the DriVLMe model architecture. DriVLMe is a multimodal Large Language Model that consists of (1) A video tokenizer that tokenize the input visual history from the CARLA [9] simulator using a frozen CLIP encoder and a linear projection layer, (2) A route planner, a tool designed to assist the LLM in finding the shortest path from the agent's current location to another landmark specified by the LLM. (3) The base large language model, which receives input in the form of video representations, situated dialogue instructions, history of physical actions, and the output planned route from the route planner. It predicts dialogue responses to human inputs and physical actions that interact with the simulator.

gation actions according to the perceptual and dialogue history. SDN is collected from human-human interactions in Wizard-of-Oz (WoZ) studies, consisting of over 8,000 utterances and 18.7 hours of control streams. In the WoZ study, a human participant engages with what they believe to be an autonomous driving agent to accomplish various navigation tasks. Behind the scenes, the actions of this agent are operated by a human wizard. This setup ensures that the participant's interactions with the agent are natural and synchronized. During the interaction, there is also an adversarial wizard who creates unexpected situations on the fly. This adversarial wizard changes environmental dynamics as well as current goals and plans by using language instructions and manipulating road conditions.

## 3.2. Problem Definitions

At time $t$, the agent is provided with a perceptual observation and a human language input, aggregated into the following model input:

- **Map knowledge.** A graph-structured topology $M$ with a list of street names $\{str_i\}$ and landmarks $\{lm_i\}$.
- **Perceptual history.** A sequence of RGB images $V = \{V_0, V_1, \cdots, V_{t-1}\}$ captured by the first-person camera. The video sampling rate is 10Hz
- **Dialogue history.** The dialogue utterances from the human ($U_{t,\text{HUM}}$) and the agent ($U_{t,\text{BOT}}$).
- **Action history.** The action history includes a sequence of previous actions $A_t = \{a_0, a_1, \cdots, a_{t-1}\}$, where each action $a_t$ is a tuple $\langle p, \alpha \rangle$ representing a physical action and its argument executed at time $t$. More details about physical action definitions are in Table 1.

The goal of the agent is to navigate to a sequence of landmarks on the map following the dialogue instructions from the human partner. To guarantee coherence in future dialogues and unforeseen events, the tasks are defined in a teacher-forcing manner. This means that during data col-

| Physical Actions | Args | Descriptions |
|---|---|---|
| LaneFollow | - | Default behaviour, follow the current lane. |
| LaneSwitch | Direction | Switch to a neighboring lane. |
| JTurn | Direction | Turn to a connecting road at a junction. |
| UTurn | - | Make a U-turn to the opposite direction. |
| Stop | - | Brake the vehicle manually. |
| Start | - | Start the vehicle manually. |
| SpeedChange | Speed ($\pm5$) | Change the desired cruise speed by 5 km/h. |
| LightChange | Light State (On/Off) | Change the front light state. |

Table 1. The high-levels action space in the SDN benchmark.

lection, the model is always presented with the actual action history $A_t$, rather than model-predicted actions during inference. The model is evaluated against the action and dialogue decisions of the human wizard. We particularly consider two sub-problems.

**The Dialogue Response for Navigation (RfN) task.** The RfN task evaluates the agent's performance in generating an adequate response in driving-related communication. At time stamp $\tau$, when the wizard makes an utterance, the agent is required to predict the dialogue response $d$. Instead of predicting only the dialogue move, we task the agent to generate the natural language.

**The Navigation from Dialogue (NfD) task.** The NfD task evaluates the agent's performance in following human instructions from dialogue. At time stamp $\tau$, when the wizard makes a decision on a physical action $\langle p, \alpha \rangle$, the agent is required to predict this physical action.

## 4. Method

### 4.1. Model Architecture

Our DriVLMe agent is a large video-language model consisting of three parts: a video tokenizer, a route planning module, and a large language model backbone. The overview architecture of DriVLMe is visualized in Figure 1.

**Video Tokenizer.** At time $t$, we can get a visual observation history $\{V_0, V_1, \cdots, V_{t-1}\}$. Given the long-range na-

ture of the SDN benchmark, we assign a window size of $T_{\max} = 40$ with step $\Delta t = 2$ to sample the vision history and form a video $V \in \mathbb{R}^{T \times H \times W \times C}$, where $H$, $W$, and $C$ are the height, width, and channel, respectively. For each video frame $V_i$, we adopt a pre-trained CLIP ViT-L/14 encoder [36] to extract the feature map $f \in \mathbb{R}^{T \times h \times w \times D}$, where $h = H/p$, $w = W/p$, $p$ is the patch size of vision transformer, and $D$ is the feature dimension of the CLIP encoder. We apply average-pooling to the feature map along the temporal dimension to get a representation $v_s \in \mathbb{R}^{(h \times w) \times D}$ and along the spatial dimensions to get a representation $v_t \in \mathbb{R}^{T \times D}$. By concatenating these two embeddings, we get the following video representation $v = \text{Concat}(v_t, v_s) \in \mathbb{R}^{(T+h \times w) \times D}$. We then use a linear projection layer $g$ to project the embedding into the language decoder's embedding space with a dimension of $K$, resulting in the final embedding $g(v) = \mathbb{R}^{(T+h \times w) \times K}$.

**LLM Backbone.** The LLM decoder is the core module that processes the input video and translates the dialogue instructions into lower-level decisions. Motivated by Video-ChatGPT [28], we adopt Vicuna-7B (v1.1) [51] as the LLM decoder. Motivated by the tool-using capability of LLMs, we introduce a planning framework for environmental understanding with the detailed prompts shown in Figure 2.

**Route Planning Module.** To enable symbolic planning for long-horizon goals, we introduce a route planner to incorporate the graph knowledge in the map $M$ into DriVLMe. The planner takes as input a given target landmark on the map $\text{lm} \in \{\text{lm}_i\}$ and the current location of the agent $l$. It then outputs a route from the agent to the target landmark following the shortest path. To call the planner, the agent can simply output $\text{Plan}(\text{lm})$. The planner returns a list of turning directions, one per intersection in the route, expressed in natural language. The final output delivered to the DriVLMe agent is a list of directional action $\{p\} = [\text{dir}_1, \text{dir}_2, \cdots]$, where $\text{dir}_i \in \{\text{left}, \text{right}, \text{straight}, \text{uturn}\}$.

## 4.2. Instruction Tuning

Motivated by Hu and Shu [15], our goal is to enhance a language model's competence as a world model and agent model by learning from embodied experiences and social interactions. The training process of DriVLMe consists of two stages: (1) the general video instruction tuning stage, focused on aligning the LLM and the video tokenizer using large-scale driving videos, and (2) the social and embodied instruction tuning stage, focused on training the LLM on the conversational data collected from real human-human dialogue and episodes of embodied experiences in a simulator.

---

> (**Video**)
> (**System Message**): You are DriVLMe. You are responsible for safely piloting a car according to the instructions of a passenger. You must communicate with the passenger and make high-level decisions regarding the current navigational goals.
> (**Prompt**): Describe what you see.
> (**LLM**, Description): I can see a car in front of me. I can only switch left lane...
> (**Dialogue & Action History**)
> (**Route Planning Instruction**): You have a planning tool that you can plan your path to the destination. You can call it by `plan(destination)`, and it will return you a plan to get to your destination. If you don't have a destination in your mind, you can return `plan(None)`.
> (**LLM**, Planning): `plan(ikea)`
> (**Route Planner**): [left, straight, ...]
> (**Prompt**): You can select a new navigational action and reply to the passenger.
> (**LLM**, Action): `SwitchLane`
> (**LLM**, Dialogue): "Ok, I will go to IKEA."

Figure 2. **Example of system message and interaction between user and DriVLMe system.** The system message is an overview of the task the agent is required to accomplish. Given the video and the observation history, the agent is required to first describe the surrounding environment, then call the planner API to plan a route to the predicted goal, and make a decision at last. The output of the LLM is highlighted.

### 4.2.1 Domain Video Instruction Tuning

Following the practice of Video-ChatGPT [28], we initialize the projection layer directly from LLaVA-7B (lightening v1.1) [24]. We adopt 50k video-text pairs from the BDD-X dataset [20] for the driving domain tuning. The pre-training images are collected from real driving videos and textual annotations of the environmental description and action explanations. We freeze the CLIP encoder and the LLM decoder, and train the projection layer only.

### 4.2.2 Social Instruction Tuning

At this stage, we used LoRA [13] to fine-tune the LLM in addition to the projector. We train the model on the whole training set of the SDN dataset, which has 13k video-dialogue pairs, including human-vehicle dialogues and long-term goals for planners. At each datapoint $\tau$, the original SDN benchmark provides the dialogue $d$ generated by human players, or physical action $\langle p, \alpha \rangle$, where $p$ is an action (e.g., `Stop`) and $\alpha$ is an argument (e.g., `left`). We aim for the agent to learn how to plan in alignment with human intentions, which involves creating a sequence of primitive actions based on the goal and dialogue history, particularly when there's a change in the goal or plan. We manually annotate plan changes based on the car's trajectory and the current dialogue. While there could be several valid paths

from the current location to the goal, we manually selected the routes that the vehicle took during the recording. These annotated plans serve as a part of the video-instruction data pairs for training, facilitating more effective learning of the planner as a tool.

### 4.2.3 Embodied Instruction Tuning

Besides the original dialogue data, we developed a data generation pipeline to obtain paired data of embodied perception and descriptions from the simulator. We replay the training sessions in the SDN benchmark to obtain the ego-centric perception, record the environmental factors such as weather and nearby objects, and then fill these details into language descriptions using templates.

- **Distance to Road End**: We compute the distance to the road's end by subtracting the current waypoint's $s$ value from the $s$ value at the road's end. The $s$ value is defined according to the OpenDrive 1.4 standard [10].
- **Lane Information**: We note the lane number the car was in, counting from the left, and record whether the car could switch to the adjacent left or right lanes.
- **Object in Front**: We identify the object directly in front of the vehicle from the ground truth obtained from the simulation, and compute the distance to it.
- **Traffic Sign Visibility**: We record all visible traffic signs (e.g., traffic lights, stop signs, speed limit signs), along with the information they displayed (red/green for lights, posted speed limits), and their distances from the vehicle.
- **Weather Conditions**: We record the current weather conditions that could impact the vehicle's control.

The text templates used to verbalize the embodied experiences are available in Appendix 8.1.

### 4.2.4 Hyper-parameters.

The input resolution of the video is set as $224 \times 224$. We use a single linear layer for projection. For the pre-training stage of the model, we trained the model for 3 epochs with a learning rate of $2e^{-5}$ and a batch size of 4. We fine-tune the LLM with LoRA [13] and ZeRO [37]. The training epoch is 2 and the batch size is 1.

## 5. Open-loop Evaluation

### 5.1. SDN Benchmark

For the open-loop evaluation, we tested the model on the test split of the SDN benchmark. The test set has two subsets, seen and unseen, where seen data points adopt either CARLA map Town01, Town03, or Town05 as the environment (which appeared in the training set). The unseen data points are from Town02, which is a relatively simple town map that was held out from training.

| Model | NfD | | RfN | | | |
|---|---|---|---|---|---|---|
| | Act↑ | Arg↑ | Move↑ | CIDEr↑ | BERT↑ | M↑ |
| **Seen Environments** | | | | | | |
| TOTO | 41.2 | 36.0 | 40.9 | - | - | - |
| GPT-4 | 53.0 | 44.2 | 11.0 | 0.06 | 0.48 | 0.09 |
| GPT-4V | 52.0 | 29.4 | 6.5 | 0.07 | 0.54 | 0.11 |
| DriVLMe | **64.2** | **74.2** | **53.8** | **0.38** | **0.73** | **0.31** |
| DriVLMe (-video) | 60.3 | 72.5 | 42.7 | 0.33 | 0.69 | 0.26 |
| DriVLMe (-planner) | 57.6 | 52.0 | 21.3 | 0.19 | 0.61 | 0.12 |
| **Unseen Environment** | | | | | | |
| TOTO | 45.8 | 41.1 | 31.0 | - | - | - |
| GPT-4 | **67.5** | 61.3 | 14.5 | 0.05 | 0.47 | 0.08 |
| GPT-4V | 63.5 | 51.6 | 7.5 | 0.07 | 0.53 | 0.13 |
| DriVLMe | 65.3 | **68.8** | **56.3** | **0.46** | **0.76** | **0.35** |
| DriVLMe (-video) | 62.6 | 68.6 | 46.5 | 0.41 | 0.73 | 0.31 |
| DriVLMe (-planner) | 58.2 | 59.1 | 23.7 | 0.22 | 0.63 | 0.13 |

Table 2. Results of open-loop evaluation on the SDN test set. The seen sessions are from CARLA map Town01, Town03, and Town05, while unseen sessions are from CARLA map Town02. The NfD task measures the agent's ability to navigate according to human instruction and the RfN task measures the agent's ability to respond to humans in a situated dialogue.

## 5.2. Evaluation Metrics

We evaluate our model on two tasks, RfN and NfD. The NfD task necessitates the agent's prediction of the physical action $\langle p, \alpha \rangle$, where $p$ represents the chosen physical action and $\alpha$ is its argument. For evaluating both the physical action and its argument, we employ accuracy metrics. In the RfN task, the agent is required to predict the dialogue output $d$. The model is tasked with predicting the dialogue move $m$ as defined in SDN. To evaluate the natural language dialogue output, we consider additional language generation metrics: CIDEr [54], BERTScore [68], and METEOR [2].

## 5.3. Baselines

**Expert Baseline.** We compared our model with TOTO [26], a baseline model implemented with an episodic transformer. Since the TOTO model does not have a text decoder and thus cannot generate dialogue, we only recorded the dialogue move prediction accuracy of TOTO.

**Generalist Baselines.** The GPT-4 and GPT-4V models are generalist LLMs we consider.[1] Due to computational constraints, rather than test both models on the entirety of the SDN test set, we chose to randomly sample data points from four strata: seen RfN, unseen RfN, seen NfD, and unseen NfD. To evaluate each model on one of these strata, we randomly sampled 200 data points and fed them into a custom prompting infrastructure similar to the structure in Table 2. For the vision-enabled model (GPT-4V), we prepended an image $V_{t-1}$ as the current visual input. To help the LLMs better understand the output format, we

---

[1]We use the OpenAI *gpt-4-0125-preview* and *gpt-4-vision-preview* models, respectively.

| Model | Description | | | Justification | | | Full | | |
|---|---|---|---|---|---|---|---|---|---|
| | CIDEr↑ | BLEU4↑ | ROUGE↑ | CIDEr↑ | BLEU4↑ | ROUGE↑ | CIDEr↑ | BLEU4↑ | ROUGE↑ |
| ADAPT | 219.35 | 33.42 | 61.83 | 94.62 | 9.95 | 32.01 | 93.66 | 17.76 | 44.32 |
| DriveGPT4 | **254.62** | **35.99** | **63.97** | 101.55 | 10.84 | 31.91 | 102.71 | 19.00 | **45.10** |
| DriVLMe | 227.05 | 33.39 | 61.02 | **132.17** | **13.39** | **33.18** | **114.16** | **19.59** | 44.83 |

| Model | Speed | | | | | Turning Angle | | | | |
|---|---|---|---|---|---|---|---|---|---|---|
| | RMSE↓ | A0.1↑ | A0.5↑ | A1↑ | A5↑ | RMSE↓ | A0.1↑ | A0.5↑ | A1↑ | A5↑ |
| ADAPT | 3.02 | 9.56 | 24.77 | 37.07 | 90.39 | 11.98 | 27.93 | 66.83 | 75.13 | 89.45 |
| DriveGPT4 | **1.30** | **30.09** | **60.88** | **79.92** | 98.44 | **8.98** | 59.23 | **72.89** | **79.59** | **95.32** |
| DriVLMe | 1.59 | 22.76 | 50.55 | 70.80 | **99.20** | 33.54 | **61.38** | 70.70 | 76.21 | 91.55 |

Table 3. Results of open-loop evaluation on the BDD-X test set. We provide evaluation results on action description, action justification, full-text generation and control signal prediction.

explain each option in the decision-making prompt. The prompt engineering details are in Appendix 8.2.

## 5.4. Main Results

As shown in Table 2, our DriveVLMe model significantly outperforms the baseline models across most metrics, except for the physical action accuracy in the NfD task for the unseen map. This discrepancy may be attributed to the unfamiliarity with the unseen Town02, though it is topographically simpler. Overall, DriVLMe can predict more precise decisions and give better responses in the situated dialogue.

## 5.5. Ablation Studies

To assess the effectiveness of various components in developing DriVLMe, we conducted an ablation study. We evaluated model performance by systematically removing specific components to observe their impact on the model's ability to generate dialogue responses and predict actions.

- **Video Input (-video)**: We removed the video processing component from DriVLMe and evaluated its performance without visual information.
- **Planner Module (-planner)**: We removed the planner module responsible for route planning in DriVLMe. This experiment aimed to assess the impact of proactive route planning on the model's navigation capabilities.

As shown in Table 2, removing the video input and the planner module both decrease the performance of the model on the RfN tasks on all metrics, indicating the contribution of both models on response generation. A similar decrease in NfD performance is observed, while the impact of removing the planner is significant, suggesting that the route planner module greatly contributes to the success of the next action prediction.

## 5.6. Evaluation on Realworld Benchmark

We also explore whether DriVLMe can transition from simulated evaluations to benchmarks involving real driving scenarios. We utilize the BDD-X [20] benchmark, which offers video clips recorded by vehicle-mounted cameras along with language interpretations and control signals. We fine-tune the DriVLMe model with LoRA for another 6 epochs on the BDD-X training set, using a learning rate of $5e^{-5}$. As indicated in Table 3, DriVLMe successfully adapts to real-world driving scenarios beyond merely navigating in a simulated environment. It outperforms the ADAPT [17] baseline and achieves comparable performance to the state-of-the-art DriveGPT4 [62] baseline, surpassing several metrics, without relying on ChatGPT-augmented data as adopted in DriveGPT4.

## 6. Closed-loop Evaluation

For the closed-loop evaluation, we developed a human-in-the-loop simulation protocol in CARLA based on the simulator developed in DOROTHIE for human studies.

## 6.1. Experimental Design

We designed our closed-loop experiment to assess the adaptability and robustness of our autonomous driving system under various dynamic scenarios. The experiment was conducted in Town01 and Town02, including both seen and unseen maps. A human subject instructed the DriVLMe agent to navigate to a preset goal by giving natural language instructions following the storyboard, and the agent attempted to follow these instructions, autonomously navigate in the environment, and communicate with the human subject. To comprehensively evaluate the system's performance, we test the model with different settings as specific in the storyboards below:

- **Long-horizon v.s. Short-horizon Instructions**: Users instruct the agent with either long-horizon instructions, involving higher-level navigational goals (e.g., "go to the KFC"), or short-horizon instructions (e.g., "turn right at the next intersection") asking for immediate maneuvers.
- **Weather Change**: A sudden weather change (e.g., rain) is triggered during driving.
- **Goal Change**: The human user asks for a change of goal

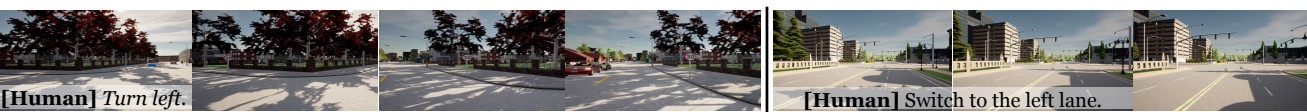

Figure 3. Examples of closed-loop evaluation of DriVLMe in CARLA, following action-level natural language instructions.

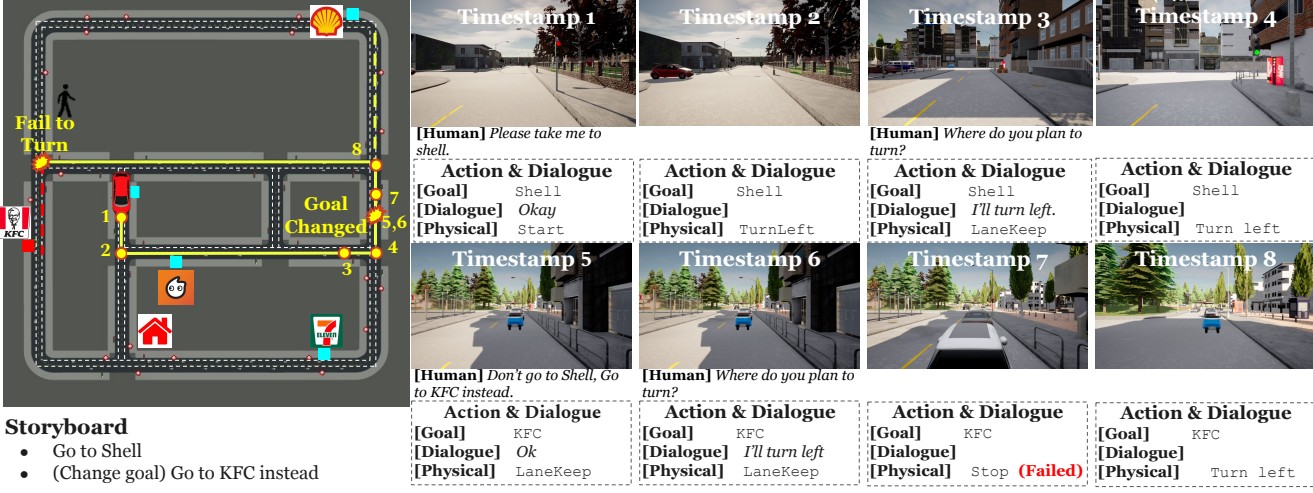

Figure 4. Example of a closed-loop evaluation session: The initial goal of the session is set to Shell, which is later changed to KFC during the course of the evaluation. The yellow solid line represents the path taken by the agent and the yellow dotted line represents the route planned by the planner. We took eight checkpoints in the whole evaluation session and recorded the input dialogue, goal prediction, dialogue response and the physical action taken for each checkpoint.

to let the agent replan the route. The human user first instructs the agent to navigate to an initial goal and then updates it.

- **Obstacle Addition**: An obstacle is placed in front of the agent to force a stop or lane change.

## 6.2. Connecting DriVLMe to Simulation

Throughout 20 pilot studies with real human subjects, agents' interactions with the simulator formed a closed-loop control mechanism. We used a local motion planner to translate the physical actions back into throttle and steering control. Due to the LLM inference rate, we limited the LLM to interact with the environment at a frequency of 2 Hz, and provided the model with the whole interaction history $H_t$ to prompt the model. For the evaluation, we used whether the final goal was achieved as the metric and recorded the failure cases for analysis.

## 6.3. Main Results

The outcomes of our experimental investigations provide compelling evidence regarding the efficacy and robustness of our proposed DriVLMe model in autonomous driving dialogue tasks, with 6 successful sessions out of 20 tests. As can be seen in Figure 3, we find that the DriVLMe model is capable of following simple human instructions and performing the physical actions as requested, in line with previous studies on foundation model agents for autonomous driving. Surprisingly, we find that DriVLMe can effectively call the route planner API for reliable graph plan-

ning and re-planning, demonstrating LLMs' tool use capabilities. The model is also robust under weather changes during the session. Still, these successful sessions are limited to cases when there is one single long-horizon goal or only one change of goal. We observe challenges with multi-turn interactions with multiple short-horizon instructions. DriVLMe also faces difficulties in handling unexpected situations and changes to environmental dynamics. Lastly, the simplified language generation from robotic experiences has triggered concerns about trustworthiness as raised by human subjects. Figure 4 shows an example of our session with a goal change instruction. We find that the agent can react to goal changes and plan turns according to the plan given by the route planner tool. However, we encountered two failure cases during the experiment. First, the agent failed to stop when the car in front suddenly stopped (timestamp 7). Second, the agent failed to predict a turn at the last intersection, causing the agent to stall at the intersection (as marked on the map). We present the video demonstration for additional details and discuss the limitations of foundation model agents in the following section.

## 7. Limitations and Future Work

Our pilot studies revealed several failure cases and technical challenges for LLM-based AD agents, outlined as follows.

**Imbalanced Embodied Experiences.** An inherent challenge in autonomous driving tasks lies in the imbalance of training data, where the majority of data points are routine

actions like lane following or maintaining a safe distance from the preceding vehicle. This imbalance can lead to model biases, particularly towards predicting more frequent actions while failing to predict actions like *stop*. Addressing this issue requires introducing robust data augmentation in embodied experiences, sampling strategies, or domain-specific knowledge into the training process to ensure comprehensive model training across diverse driving scenarios.

**Limited World Modeling and Visual Understanding.** Our experiment revealed instances where the visual encoder failed to capture critical world states due to low image input resolution, such as the color of traffic lights or the interpretation of traffic signs. The absence of optical character recognition (OCR) capabilities further exacerbates the risk of misinterpreting traffic signs and thus breaking traffic rules. Future efforts could explore techniques to enhance image resolution, integrate OCR functionalities, or incorporate complementary sensor modalities to enrich perception and improve overall world modeling performance.

**Unexpected Situations and World Dynamics** Our closed-loop experiment results on unexpected situations like encountering an obstacle have revealed limitations in the LLM agent's ability to effectively address out-of-distribution corner cases. Such cases are common in real-world driving scenarios, highlighting the need for enhanced capabilities in LLM-based autonomous driving agents to handle unforeseen circumstances. One potential direction for the future is to enable agents to learn from in-the-wild driving video/data and develop a better world model. Alternatively, integrating LLMs with preset knowledge about appropriate responses to unexpected situations could also be beneficial.

**Language Generation from Embodied Experiences** Furthermore, our investigation revealed that the language generated by our model tends to be oversimplified, primarily consisting of straightforward responses to human instructions or simplistic yes/no replies. Additionally, the model cannot initiate a dialogue with a human instructor, e.g., requesting additional advice or low-level instructions. Future work should focus on enhancing the model's conversational initiative, enabling self-motivated dialogue.

**Multi-turn Interactions and Instruction Following** Our closed-loop experiments also suggest the challenges of multi-turn interactions and instruction following. As the conversation goes on, the agent occasionally fails to retain previous long-horizon instructions, leading to wrong goal predictions and subsequent disruptions to the planning route. This issue underscores the critical importance of memory retention and context awareness in maintaining an agent model, particularly in situations where extensive dialogue exchange happens. Addressing these challenges through the implementation of memory-based mechanisms

within LLM architectures or adding some memory modules in the autonomous driving agent framework could significantly enhance the agent's ability to follow complex instructions in a complex environment that needs lots of human-agent collaboration.

**Limited Theory of Mind and Trust-worthiness** Another critical limitation observed in our study is the absence of a situated Theory of Mind (ToM) [27] in the autonomous agent. At times, the agent misinterprets the instructor's intentions, mistakenly perceiving low-level instructions as cues to abandon the previously provided long-horizon instruction and predict the goal incorrectly. The agent fails to recognize that the instruction may simply be specifying details within the ongoing long-horizon instructions. This highlights the need for autonomous driving agents with a nuanced understanding of the instructor's intentions and context, enabling better agent modeling for their interaction partners, thus, gaining trust from humans in real-world navigation tasks.

**Unacceptable Inference Time** Our model's single inference time takes approximately 5 seconds, which significantly exceeds the interval between two decision points, posing a substantial challenge in real-world scenarios where rapid decision-making is imperative. While this delay is avoidable in a simulated environment through step-by-step simulation, addressing this inference time disparity is crucial for practical deployment. Future research directions may focus on distilling the model, leveraging hardware acceleration, or implementing efficient inference strategies to mitigate this bottleneck. This also raises a research problem of balancing the length of the Chain-of-Thought reasoning to reduce the inference time while keeping a comparable performance in task accomplishment.

## 8. Conclusion

In this work, we presented DriVLMe, an LLM-based autonomous driving agent that leverages both embodied experiences in a simulated environment and social experiences in real human dialogue. The egocentric perception and conversational interaction empower DriVLMe to engage in meaningful dialogues with human passengers while navigating complex driving environments. Through empirical evaluations, we demonstrated the effectiveness and versatility of DriVLMe in autonomous driving dialogue tasks, showcasing significant improvements in both physical action prediction and dialogue response generation metrics. Our findings have demonstrated the potential of DriVLMe in enabling human-agent communication and autonomous driving, and on the other hand, reveal ed several key limitations and challenges. of foundation models as AD agents, highlighting areas that need future enhancement.

# Acknowledgment

This work was supported by the Automotive Research Center (ARC) at the University of Michigan and NSF IIS1949634. The authors would like to thank the reviewers for their valuable feedback.

# Ethics Statement

The institution's Institutional Review Board (IRB) considered this project exempt from ongoing review, registered under eResearch ID HUM00205133. The SDN benchmark includes human-generated speech and demonstrations, while the BDD-X dataset features human-generated annotations. Our use of both datasets is in compliance with their licenses and exclusively for research purposes.

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

# Appendix

## 8.1. Language Templates for Verbalizing the Embodied Experiences.

With the data about the surrounding environment, we use templates to generate synthetic data as the caption of the input video:

- **Distance and Turning Decisions**: For the distance to the road end, we generated different outputs based on the distance recorded. When the distance is larger than 10, we used the prompt "I am far from the end of the road. I don't need to make a decision for turning now." When the distance is larger than 5 while smaller than 10, we used the prompt "I am near the end of the road. I don't need to make a decision for turning now." When the distance is smaller than 5, we used the prompt: "I am at the end of the road, I need to stop if there is a red light, or make a decision to turn left, turn right, or go straight now."

- **Lane and Lane Switching Decisions**: For the lane information, we used the prompt "I'm on the {lane_number} lane from the left of the road", and based on whether a lane change is affordable, we chose from the 4 prompts: "I'm not able to change lane", "I'm only able to change to the right lane", "I'm only able to change to the left lane", "I'm able to change to both right and left lane."

- **Object and Stop Decisions**: For each object in front, we used the template "There is a obstacle {object_type} in front of me, the distance is {distance}." For the object type, we used the object class in CARLA (e.g. vehicle, pedestrian, traffic sign).

- **Signs and Stop Decisions**: For each traffic sign in front, we used the template "There is a {sign_name} that is {distance} meters from me, showing {state}." The sign_name is the name of the sign while the state is the information the sign displayed (e.g., red/green for lights, posted speed limits).

- **Weather**: For the weather, we straightly described that using the template "It's {weather}."

## 8.2. Prompt Engineering for GPT-4 Baseline

Each prompt template we used for the GPT-4 baseline consists of the following components in order:

1. **Image:** For the vision-enabled model only (GPT-4V, not GPT-4), we prepended an image of the third-person driver view.

2. **Header:** Informs GPT that it must act as a Chauffeur, piloting a car while talking with its passenger.

3. **Dialogue History:** Turn-by-turn record of the conversation between passenger and driver prior to the time of prompting.

4. **Current Map:** A text-based representation displaying the map along with landmarks, street names, and the vehicle location

5. **Physical Action History:** Turn-by-turn record of the previous physical actions taken by the driver.

6. **Planner:** Asks GPT to call a planning module using the form plan(landmark). If GPT both uses this API correctly and selects the correct landmark, the planning module provides the plan (a sequence of turns at each intersection).

7. **Question 1:** For NfD, this segment asks GPT a multiple-choice navigational question. For RfN, it asks GPT what type of dialogue it would like to output.

8. **Question 2:** For NfD, if the correct action takes an argument (e.g., for turning, the argument is a direction), this segment asks for the argument in a multiple-choice format. For RfN, this segment asks for the natural language dialogue. For question 2, we utilize teacher forcing, providing the GPT model with the correct answer to question 1 even if it is answered incorrectly.

