# OpenReview forum: "DriVLMe: Enhancing LLM-based Autonomous Driving Agents with Embodied and Social Experiences"
_thecvf.com/CVPR/2024/Workshop/VLADR — VLADR 2024 Poster_

### Official Review · Reviewer_ksKt · 2024-04-21
**Good paper**

**Rating:** 6
**Confidence:** 5

**Review:**

This paper proposes a vision language model based agent for autonomous driving. The overall architecture is similar to existing work, however, this paper performs closed-loop simulation on Carla, which is novel. Also, the LoRA fine-tuning is more efficient compared to prior works such as GPT-Driver. Therefore, I recommend acceptance of this paper. One related work that is worth comparing to is Agent-Driver [1].

[1] A Language Agent for Autonomous Driving. Mao et al.

---

### Decision · Program_Chairs · 2024-04-22

Accept (Poster)